# Exposures to IARC Carcinogenic Agents in Work Settings Not Traditionally Associated with Sinonasal Cancer Risk: The Experience of the Italian National Sinonasal Cancer Registry

**DOI:** 10.3390/ijerph182312593

**Published:** 2021-11-29

**Authors:** Alessandra Binazzi, Carolina Mensi, Lucia Miligi, Davide Di Marzio, Jana Zajacova, Paolo Galli, Angela Camagni, Roberto Calisti, Anna Balestri, Stefano Murano, Sara Piro, Angelo d’Errico, Matteo Bonzini, Stefania Massacesi, Denise Sorasio, Alessandro Marinaccio

**Affiliations:** 1Department of Occupational and Environmental Medicine, Epidemiology and Hygiene, Istituto Nazionale per l’Assicurazione Contro gli Infortuni sul Lavoro, 00100 Roma, Italy; d.dimarzio@inail.it (D.D.M.); a.marinaccio@inail.it (A.M.); 2Sinonasal Cancer Registry of Lombardy, Epidemiology Unit, Fondazione IRCCS Ca’ Granda Ospedale Maggiore Policlinico, 20100 Milano, Italy; carolina.mensi@unimi.it (C.M.); matteo.bonzini@unimi.it (M.B.); 3Sinonasal Cancer Registry of Tuscany, Occupational and Environmental Epidemiology Unit, Institute for Cancer Research, Prevention and Clinical Network (ISPRO), 50139 Firenze, Italy; l.miligi@ispro.toscana.it (L.M.); s.piro@ispro.toscana.it (S.P.); 4Sinonasal Cancer Registry of Piedmont, Occupational Health and Safety Department, CN1 Local Health Authority, 12037 Saluzzo, Italy; jana.zajacova@aslcn1.it (J.Z.); denise.sorasio@aslcn1.it (D.S.); 5Sinonasal Cancer Registry of Emilia Romagna, Occupational Safety and Prevention Unit, Public Health Department, Bologna Local Health Authority, 40121 Bologna, Italy; paolo.galli@ausl.bologna.it (P.G.); angela.camagni@ausl.bologna.it (A.C.); 6Sinonasal Cancer Registry of Marche, Department of Prevention, Unit of Workplace Prevention and Safety and of Occupational Epidemiology (SPreSAL Epi Occ), Regional Health Authority Marche, 62012 Civitanova Marche, Italy; roberto.calisti@sanita.marche.it (R.C.); stefania.massacesi@sanita.marche.it (S.M.); 7Sinonasal Cancer Registry of Lazio, Department of Epidemiology, Servizio Sanitario Regionale del Lazio, 00100 Roma, Italy; anna.balestri@asl.vt.it; 8Sinonasal Cancer Registry of Autonomous Province of Bolzano, Alto Adige Health Authority, Occupational Medicine Unit, 39100 Bolzano, Italy; stefano.murano@sabes.it; 9Epidemiology, Local Health Unit ASL TO3, Piedmont Region, 10095 Grugliasco, Italy; angelo.derrico@epi.piemonte.it; 10Department of Clinical Science and Community Health, University of Milano, 20100 Milano, Italy

**Keywords:** sinonasal cancer, occupational exposure, occupational cancer, epidemiological surveillance

## Abstract

The aim of this study is to highlight tasks and jobs not commonly considered at high risk for sinonasal cancer (SNC) identified by Regional Operating Centers currently active in the Italian National Sinonasal Cancer Registry (ReNaTuNS), which retrieve occupational histories through a standardized questionnaire. Data on exposures to IARC carcinogenic agents in work settings unknown to be associated with SNC risk were collected and analyzed. Out of 2,208 SNC cases recorded in the ReNaTuNS database, 216 cases and their worked exposure periods were analyzed. Unsuspected jobs with exposure to wood dust include construction-related tasks, production of resins, agriculture and livestock jobs (straw and sawdust), and heel factory work (cork dust). Other examples are hairdressers, bakers (formaldehyde), dressmakers, technical assistants, wool and artificial fiber spinners, and upholsterers (textile dusts). Moreover, settings with coexposure to different agents (e.g., wood with leather dusts and chromium–nickel compounds) were recognized. The study describes jobs where the existence of carcinogenic agents associated with SNC risk is unexpected or not resulting among primary materials employed. The systematic epidemiological surveillance of all epithelial SNC cases with a detailed collection of their work history, as performed by a dedicated population registry, is essential for detecting all potential occupational cases and should be considered in the context of forensic medicine and the compensation process.

## 1. Introduction

Cancers of the nasal cavity and paranasal sinuses are rare malignancies, accounting for 3% to 5% of all head and neck cancers and less than 1% of all cancers [1]. Epithelial tumors represent more than 80% of all sinonasal tumors, with “sinonasal squamous cell carcinoma” (mainly occurring in the maxillary sinus and nasal cavity) and “intestinal-type adenocarcinoma” (which principally arises in the ethmoid sinus) being the most common subtypes. Sinonasal malignancies have an estimated incidence rate ranging from 0.5 to 1 case per 100,000 people and a 5-year overall survival reported to be at 30% to 50% [2].

Exposure linked to occupational settings is a major risk factor for sinonasal cancer (SNC), which is occupational exposure. Despite the low absolute risk in the general population, SNC is associated with high relative risks for specific chemical exposures and workplaces. The occupation-attributable fraction is estimated to be in the range of 20–46% and is particularly elevated for adenocarcinomas (77%) [3,4].

According to the International Agency for Research on Cancer (IARC), carcinogenic agents associated with the nasal cavity and paranasal sinuses include: wood dust, leather dust, nickel compounds, radium-226 and its decay products, radium-228 and its decay products, isopropyl alcohol during production, tobacco during smoking (with sufficient evidence of carcinogenicity in humans), dust in carpentry/joinery, hexavalent chromium compounds, formaldehyde, and textile during manufacturing (with limited evidence) [5].

This strong relationship between SNC incidence and exposure to specific occupational carcinogens supports the development of a specific epidemiological surveillance system that in Italy is compulsory since 2008, through the “National Sinonasal Cancer Registry” (Registro Nazionale Tumori Naso-Sinusali (ReNaTuNS)). This is a nationwide cancer registry coordinated by the Italian Workers’ Compensation Authority (Istituto Nazionale per l’Assicurazione contro gli Infortuni sul Lavoro (INAIL)) aimed at gathering the clinical characteristics of SNC cases, identifying those of occupational origin and estimating the incidence rates of SNC [6].

The ReNaTuNS has a regional structure with local registries, established at the Regional Operating Centers (Centri Operativi Regionali (COR)), dedicated to the active search for cases in health-care centers and the assessment of previous exposure to occupational carcinogenic agents.

Currently, the registry covers a portion of the national territory (27%), around 30 million inhabitants.

The main strength of the ReNaTuNS is represented by the analytical investigation of the occupational exposure history of each collected patient, in some cases leading to the identification of exposures not usually associated with SNC, which may ultimately result in compensation for work-related illness and the implementation of preventive measures [7,8].

By using data on SNC cases collected by CORs, the purpose of this paper is to describe economic sectors and jobs not conventionally considered to be at high risk for SNC, but where a definite occupational exposure to IARC carcinogens for SNC has been assessed and recognized.

## 2. Materials and Methods

The CORs connect the local archives that identify SNC cases to the centralized ReNaTuNS, and warrant the quality of collected information. The CORs obtain documentation about each case from departments that provide diagnosis and treatment for SNC (pathology, otolaryngology, maxillofacial surgery, radiotherapy wards) that make their clinical records available. Moreover, they carry out an active search of incident SNC cases from the same departments. The completeness of the registration is regularly checked by surveys using current regional health data sources (e.g., hospital discharge forms, mortality registries).

Currently, the registry covers around 50% of the Italian population. The CORs are established in the Italian regions of Piedmont, Lombardy, Autonomous Provinces of Trento and Bolzano, Emilia-Romagna, Tuscany, Marche, and Lazio.

Diagnosis of SNC (ICD-10 codes C31.0 to C31.9) consists of either confirmed (verified by histological examination) or probable cases (without histological confirmation but with clinical diagnosis and radiologic check through CT or MRI). Histological types are defined according to the WHO 2017 classification [1]. National guidelines for the keeping of the registry guarantee data quality by means of criteria that specify the coherence between the level of diagnostic certainty and morphology as well as the correspondence between morphology and the anatomical site of SNC [9].

For the aims of the present study, only confirmed cases were selected, taking into account the histological type of the tumors.

The exposure assessment is primarily based on the information derived from a standardized questionnaire administered to patients or their next of kin by trained interviewers, which allows the evaluation of lifetime occupational history (including industrial sectors, plants, jobs, and specific tasks performed) and exposures in extra-occupational settings (domestic or during hobby activities). Considering occupational exposures, data obtained with a job-specific questionnaire (forestry, agriculture, wood industry, tannery industry, and shoe factory) were used; the job-specific questionnaires collect detailed information about the work setting and tasks performed. The approach used in this study was to assess the information derived from the questionnaires with subsequent ratings to assign exposure to certain or suspected carcinogen agents for SNC. Industrial hygienists or experts from each geographic area examined the information collected and assessed the level of probability.

According to the national guidelines [9], when exposure to a carcinogen agent occurs in a workplace, subjects are classified to have “occupational exposure”; otherwise, extra-work settings (environmental, domestic) or activities (hobbies) imply a nonoccupational exposure.

Each occupational history of SNC cases is coded using a standardized classification of industrial sectors or job titles that allows for performing analyses by economic sector and jobbing [10]. A level of exposure to each carcinogen agent for TUNS is attributed to each work period (in terms of probability of exposure).

Occupational exposure classification in terms of probability of exposure is qualitative: it can be definite, probable, or possible. Definite occupational exposure is assessed for subjects whose work has involved exposure to a causal agent: such exposure must be documented by a declaration of the interviewed patient or environmental investigations, reports from supervisory boards, company documentation, or statements from colleagues/employer/patients’ relatives.

Probable occupational exposure is attributed to subjects who have worked in a company where a causal agent was certainly used but whose exposure cannot be reliably documented/evaluated due to deficient or incoherent information collected in the questionnaire.

Lastly, subjects who have worked in a company belonging to an economic sector where exposure to a causal agent may have occurred but where information supporting such exposure is insufficient have a possible occupational exposure.

SNC cases with an identical level of exposure either attributed to different economic sectors or because of differing causal carcinogenic agents are assigned multiple exposures (i.e., more than one exposure for each subject). Therefore, the number of exposures can be equal or higher than the number of exposed subjects, because more than one work period is considered in the exposure history of each SNC case.

In this paper, we present an analysis of retrieved occupational exposures to IARC carcinogenic agents for the nasal cavity and paranasal sinuses (as specified above) in subjects employed in work settings not traditionally associated with SNC. 

## 3. Results

From 1996 to 2019 in the ReNaTuNS database, 2208 diagnosed SNC cases were recorded. Out of these, the CORs selected 216 cases featuring a definite diagnosis of primitive tumor of the nasal cavities or paranasal sinuses and occupational exposure to an IARC carcinogen, with the nasal cavities or paranasal sinuses as target organs, which occurred exclusively in unexpected work sectors (i.e., unknown for exposure to such carcinogens).

The number of exposure periods worked was 240 (each subject may have been exposed in more than one work or job sector). The identified IARC carcinogens in exposure histories included wood dust, leather dust, chromium, nickel, formaldehyde, and textile dusts. The largest number of exposures was observed in men for all carcinogenic agents, except for textile dust, with a slightly higher percentage for women (52.9%). The mean age at diagnosis ranged from 62.0 (±11.9) to 69.2 (±9.2) in the groups exposed to formaldehyde and leather dust, respectively. Attribution to occupational exposure was definite for most of the agents, except for textile dust and formaldehyde (probability was more frequent: 52.9% and 48.5%, respectively) (Table 1).

Nasal cavities were the most frequent tumor sites, followed by maxillary and ethmoid sinuses. The distribution of histological types highlighted adenocarcinoma as the most frequent in exposures to wood dust, textile dust and leather dust, and squamous cell carcinoma in exposures to formaldehyde, chromium and nickel (Figure 1).

In Table 2, a description of atypical work situations, where exposure to carcinogenic agents were assessed by industrial hygienists in each COR, is presented. For each carcinogen, we report the job sector and context of the retrieved cases (Table 2). With regard to wood dust, exposures are divided into subgroups: generic wood dusts, straw, sawdust, and cork dust. We found jobs or tasks already confirmed to be at risk for SNC (woodworker, manufacture of wooden objects or pruning, and forestry, which are not reported in Table 2), but also others such as framer, construction-related jobs, production of resins, and wooden mold production in the steel industry, which are so far unknown. Some exposures to straw and sawdust were reported in agriculture and livestock jobs and in poultry breeding, where saw litters were employed, as well as in binding up chairs or glass vessels with straw. A relevant number of cork dust exposures were detected, mainly in cork production and heel factory worker. Exposure to leather dust was found in some specific tasks related to shoe and bag production. SNC cases occurred also in several sectors involving the use of chrome paints (e.g., plating or welding) and hexavalent chromium (Cr(VI)) in the cases of a designer and a photoengraver. Exposure to formaldehyde was observed in already-known jobs, such as generic woodworker (not reported in Table 2), but also hairdresser and barber, pastry chef and baker, and many other economic sectors of plastics, metals, textiles, chemical processing. Exposure to textile dusts was detected mainly among dressmakers, technical assistants or mechanical textile workers, wool and artificial fiber spinners, upholsterers, loom operators, and knitters.

Occurrence of exposure to IARC carcinogenic agents for the nasal cavity and paranasal sinus may coexist in the same work period of an SNC case, as reported in Table 3. Coexposures to wood and leather dusts were observed in the treatment of fur and leather and in cork use for sole production. Exposure to wood dust was found associated also with textile dusts, chromium, welding fumes, and formaldehyde, while exposure to leather dust with nickel and chromium. Several activities involved exposure to chromium and nickel and to formaldehyde along with textile dusts.

## 4. Discussion

Exposure to recognized carcinogens and other causal agents in SNC cases was explored in a previous study in the ReNaTuNS and evidenced occupational exposure in 63% of the registered SNC cases (73% in men, 35% in women) [6].

The percentage of interviews was high (more than 75% in the ReNaTuNS dataset), with detailed information on occupational exposure captured through the questionnaire.

The present study highlights several tasks or jobs with a definite or possible exposure to carcinogens in a substantial number of SNC cases, where association with occupational exposures to carcinogenic agents is not expected.

Economic activities where the presence of carcinogens could have been unknown, or not resulting from use of primary materials, have emerged, suggesting the need for a strict control of exposure to these hazards.

This is a valuable result of Italian epidemiological surveillance: thanks to a detailed investigation of the work history of each recorded SNC case, it has identified unusual circumstances of exposures for SNC, with important consequences in planning preventive measures.

Occupational exposure to each IARC agent associated with SNC is discussed below.

### 4.1. IARC Carcinogenic Agent

#### 4.1.1. Wood Dust

In the present study, wood was observed as primary material in unexpected jobs for this kind of exposure (e.g., architect, surveyor, bricklayer, house painter). In particular, house painters can be exposed to paint dust particles from mechanical abrasion (i.e., sanding) of coated wood surfaces; bricklayers in using wooden planks or beams to create scaffolding before installing building components, such as roofs, or before pouring concrete; and architects in attending the installation of parquet and wood paneling. A meta-analysis estimated a 62% excess risk of SNC for previous work in the construction industry, and wood dust appears to be the agent mainly responsible for this association [11]

#### 4.1.2. Sawdust

We found evidence of occupational exposure to straw and sawdust in agriculture and poultry breeding associated with the use of saw litters. Studies investigating the effect of organic dust exposures among farmworkers have suggested that handling and spreading straw or hay is a strong predictor of inhalable dust exposure and risk of respiratory disease [8,12]. Particularly, in a turkey breeding shed, concentrations in low dust exposure jobs and during litter replacement exceeded the threshold limits [7]. Apart from the farm sector, we found sawdust exposure in jobs such as butcher or printer, as well as in jobs in metal furniture and loading dock in harbors.

#### 4.1.3. Straw

Exposure to straw in pressing and in binding up vessels and chairs in straw was detected in some SNC cases of the present study. In a population-based case–control study of cancer of the nasal cavity and sinuses conducted in Shanghai, the use of wood and straw as cooking fuel was linked to moderate increases in risk of squamous cell carcinoma [13].

#### 4.1.4. Cork Dust

We found several cases of exposure to cork dust, mainly in the industrial processing of cork and footwear production. Cork is obtained from the bark of a cork oak tree (*Quercus suber*), a medium-sized tree with a very bulky bark, which is found principally in the Mediterranean area. Italy is a leader in the production, processing, and trade of cork. The exact composition of cork is not yet well known, although tannins can be extracted from it [14]. The ultimate carcinogens in wood are unknown, but, for example, extractives and tannins (phenolic compounds) have been suspected [15]. Solid wood and bark contain many similar compounds whose concentrations differ only quantitatively. In a CAREX project, workers exposed to bark are exposed to wood dust mainly on this chemical basis [16]. Suberosis is a recognized form of pneumoconiosis that affects cork workers, while only recently, SNC cases were observed (prevalently intestinal-type adenocarcinomas) among subjects occupationally exposed to cork dust [17,18,19,20].

#### 4.1.5. Leather Dust

We observed anomalous exposures to leather dust of an office employee and in specific jobs, such as gluer and leather miller for heel covering and upper edger. In Italy, several studies have investigated occupational exposure to leather dust in shoe factories, a well-developed industrial sector, where a carcinogenic risk can be attributable to workers engaged in the whole process of the shoe industry [21]. Concerning tannery workers, in a study of workers employed in making leather tanned with vegetable extracts, one death from nasal cancer was reported among men who worked with sole and heel leather [22]. A series of eight cases of tannery workers were also observed in an area in the Tuscany region characterized by the presence of numerous tannery industries [15].

#### 4.1.6. Textile Dust

We observed exposure to textile dusts in a variety of jobs, ranging from dressmaker to technical assistant or mechanical textile worker, from wool and artificial fiber spinner to upholsterer or loom operator or knitter. After wood dust, these were the most numerous exposures observed. Textile dusts are an established risk factor for sinonasal malignancies, particularly associated with adenocarcinomas and other epithelial neoplasms [4,6,11,23]. Similar results were found in the present study (47% and 14%, respectively). A meta-analysis of epidemiological studies on textile industry workers evidenced increased risks among workers exposed to cotton dust and among workers involved in spinning or weaving [24]. Furthermore, in a previous study an excess of SNC was found among fiber preparers, bleachers, weavers, dyers, and textile product finishers [25].

#### 4.1.7. Chromium

In our study, we have described some SNC cases in sectors involving the use of chrome paints, as well as the use of hexavalent chromium in a designer (as chromic acid for photo developing) and in a photoengraver.

Moreover, we detected exposure in the chrome plating of rollers for punch cards: a comparable circumstance was described in an SNC case with exposure during the chrome plating of magnesium cylinders to obtain punched cards for computer data processing [26].

#### 4.1.8. Formaldehyde

We found exposures to formaldehyde among bakers, pastry workers and chefs. Samples taken from the pastry shop of a bus terminal area showed significant concentrations of formaldehyde, although acetaldehyde predominated. The authors suggest that such results depend on the activities carried out inside the pastry shop, related to food preparation [27]. In this regard, a recent hypothesis is that formaldehyde and acetaldehyde exposure occurs in bakeries and pastry industries particularly in processes that involve a strong leavening [28].

Other exposures to formaldehyde in this study were observed in tire production finishing. A study measured the airborne concentration of formaldehyde in tire manufacturing and reported the highest concentration in the compounding process [29].

Moreover, formaldehyde has been found to off-gas from flint sandpaper containing urea–formaldehyde resin as a minor component in a double glue system [30].

### 4.2. Other Tasks or Jobs: Hairdressers

In the present study, a significant number of SNC cases were found among hairdressers and barbers. Hairdressers may be exposed to volatile solvents, propellants, and aerosols from hair sprays, as well as to formaldehyde, methacrylates, and nitrosamines contained in many hair care products [31]. The IARC recently reaffirmed that the occupational exposure of hairdressers and barbers is “probably carcinogenic” and that, globally, there is “limited evidence” on this carcinogenicity [32].

A meta-analysis found an increased risk in several anatomic sites, from around 30% for lung and bladder cancers to 50–60% for multiple myeloma and larynx cancer [33]. The Occupational Safety and Health Administration (OSHA) found that the average concentrations of formaldehyde in three hairdressing salons while using hair straightening products were above OSHA’s short-term exposure limit (STEL). Additionally, the maximum formaldehyde concentration measured during the blow drying phase was five times higher than the STEL. The results showed that high temperature can make the ingredients of products vaporize and lead to significant exposures. From an OSHA’s survey, certain hair straightening products were found to contain formaldehyde in the range of 0.01–11.8%. Moreover, formaldehyde releasers, such as methyl glycol, which can slowly release formaldehyde, were also detected [34].

### 4.3. Coexposure to Carcinogenic Agents

In the present study, several SNC cases were identified in occupational settings with coexposure to different carcinogenic agents.

Exposure to sawdust was observed in tasks concerning drying operations of fur and leather during skin and fur processing. The use of sawdust is documented in the treatment of tanned pelts with an oil solution, next cleaned in rotating drums containing sawdust, which absorbs moisture and excess oil [35].

In our study, the presence of wood dust was associated with exposure to textile dusts in the laying and painting of wooden floors; with exposure to chromium in engineering laboratory activities; with exposure to welding fumes in coffin production at a funeral home, and with exposure to formaldehyde in livestock farming. Moreover, exposure to cork dust was found associated with exposure to leather dust in cork sole production and manufacturing.

In a sales agent who was a designer of models and a seller of leather garments at tanneries, we found coexposure to leather dust and chromium. Employees engaged in the chrome tanning and finishing of leather were studied for potential exposure to hexavalent chromium salts: trivalent basic chromic sulfate is used to produce softer, thinner leathers for handbags, gloves, garments, upholstery, and the upper parts of shoes [36].

Exposure to both textile dusts and formaldehyde was found in a loom operator in a woolen mill and spinning mill, and in an apprentice at a tailor’s shop. An Indian study reported an obstructive pattern of respiratory changes in workers in the woolen industry [37]. Vapors of solvents and other substances, including formaldehyde, have been reported in ventilation exhausts of textile manufacturing plants due to the gradual vaporization of formaldehyde and residual solvents in carpets and fabrics used for upholstery and curtains [38].

Finally, it is noteworthy mentioning the frequent overlapping of chromium with nickel exposure: in this study, the professions involved included graphic designer, nickel and chrome paint producer, and some jobs in the steel industry, in the anodic oxidation of aluminum, and in the use of chrome and nickel paints. An increased risk of SNC from nickel/chromium exposure was estimated in a meta-analysis [11]. SNC was associated with nickel exposure also in a recent Italian case–control study, but it was not possible to estimate a risk due to the absence of exposed controls [24].

### 4.4. Tumor Morphology

In the present study, adenocarcinoma was confirmed as the most frequent histotype for exposures to wood dust, leather dust, and textile dust, as reported in other studies [3,11,24]. A high prevalence of squamous cell carcinoma was found for exposures to nickel, formaldehyde, and chromium. The relationship between squamous cell carcinoma and occupational exposures is yet to be defined, and epidemiological data are scarce. However, in a recent case–control study, excess risks were found for ever exposure to welding fumes, organic solvents, hexavalent chromium (Cr(VI)), leather dust, and nickel [24]. In previous studies, the most reported cases of chromium-induced nasal cancer were squamous cell carcinoma [27,39].

With regard to other epithelial neoplasms, we found these more frequently in subjects exposed to textile dusts and formaldehyde, while neuroendocrine tumors appeared in a very small number and were equally distributed among exposures (except for leather dust and formaldehyde).

### 4.5. Strengths and Limitations of the Work

SNC has a high occupational attributable fraction, and in this context, the unicity and strength of the Italian registry is the availability of a detailed occupational history for each included SNC case, with a peculiar description of the task performed. This feature gives reason to the higher number of cases recorded by the ReNaTuNS, with respect to the insurance notifications of occupational SNC cases to the National Institute for Insurance against Accidents at Work (INAIL): the procedure of finding evidence of patients’ exposure history for compensation purposes is not easy in occupational medicine. Rather, a registry, such as the ReNaTuNS, makes available a comprehensive investigation of occupational exposures at the individual level. In this context, the SNC surveillance system becomes an important tool to support and improve the efficiency of the compensation system.

The Italian registry shows how SNC surveillance not only identifies recognized set-tings of exposure, but also may detect conditions of unknown exposure relevant for prevention: in fact, an unexpected spectrum of activities involved in the use of substances carcinogenic for the nasal cavity and paranasal sinus emerged from this study.

Anyway, the exposure assessment was qualitative, and the ability to effectively identify the modalities of carcinogen exposure cannot be fully consistent among regional registries despite the use of a shared structured questionnaire.

This study has to be considered as a large case series providing a detailed catalogue of situations in which SNC cases have been exposed to carcinogens, which are known or suspected to cause these tumors, in sectors and jobs where they would have not been expected. In our view, this work may be relevant in guiding public health practitioners in the identification of the occupational etiological agents of these tumors, suggesting possible sources of exposure to these carcinogens in industries and jobs usually not considered at high risk for SNCs.

Furthermore, it must be outlined how the identification of causal exposure can be hard due to the potential interaction of more agents: for this reason, quantitative measures of exposure assessment through analytical studies should be taken.

So far, for SNCs among hairdressers, the identification of the causal agent has remained difficult, and only hypotheses could be assumed.

Until the recent updating of the national guidelines, the Italian SNC registration system has suffered some limitations. At first, differing modalities of data collection have persisted at the local level up to now, leading to a nonhomogeneous coverage of data collection at the national level. Then, another point is the complexity of SNC diagnosis, particularly in the identification of the specific site of origin of large sinonasal tumors. Finally, because SNC is an invasive tumor that often prevents patients from getting any information, difficulties in exposure assessment exist.

The national guidelines for the keeping of the registry, which has recently been updated, will contribute to improving the validity of data [9]. The availability of proper data is essential for planning epidemiological studies, also in collaboration with other networks of rare cancers. The surveillance of SNC cases through a dedicated cancer registry is a strong epidemiological tool that could be implemented also in other countries, allowing both the identification and the compensation of this occupational disease.

## 5. Conclusions

Despite improvements in the surveillance of patients, SNC is still characterized by high mortality and poor quality of life among survivors. With its very high occupational attributable fraction, the epidemiological surveillance of SNC cases is crucial, considering the high number of workers often unaware of the risk factors they are exposed to. Without a detailed investigation on work history, such as that obtained through a dedicated questionnaire administered by trained interviewers, all the SNC cases identified in this study would not have been identified as occupational, nor would they have been compensated.

A recognition of hazardous exposures in an unexpected scenario and a successful lowering of occupational risks could reduce the incidence of this disease and ensure compensation for work-related illness.

## Figures and Tables

**Figure 1 ijerph-18-12593-f001:**
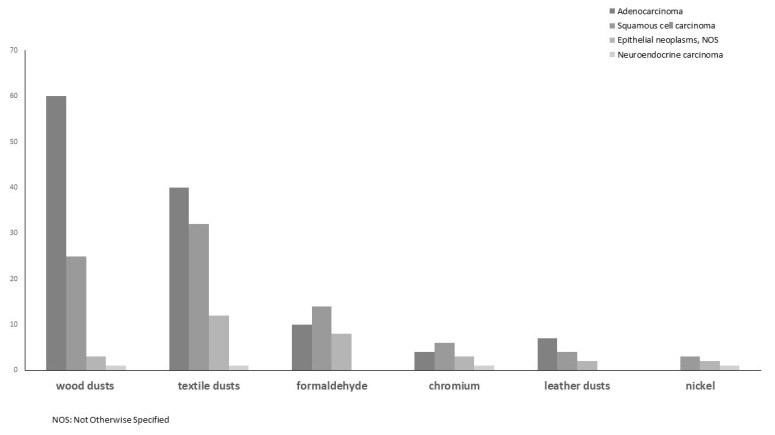
Distribution of SNC morphology (%) by exposures to IARC carcinogenic agents for nasal cavities and paranasal sinuses.

**Table 1 ijerph-18-12593-t001:** Exposures to IARC carcinogenic agents for nasal cavities and paranasal sinuses (*n*, %) by gender, age, and exposure evaluation.

Carcinogenic Agent	All	Males	Females	Age	Probability of Exposure		
					Definite	Probable	Possible
	*n* (%)	*n* (%)	*n* (%)	Mean ± SD	*n* (%)	*n* (%)	*n* (%)
**Wood dust**	89	65 (73.0)	24 (27.0)	66.6 (±10.5)	75 (84.3)	4 (4.5)	10 (11.2)
**Leather dust**	13	7 (53.8)	6 (46.2)	69.2 (±9.2)	11 (84.6)	1 (7.7)	1 (7.7)
**Chromium**	14	13 (92.9)	1 (7.1)	64.1 (±10.1)	12 (85.7)	1 (7.1)	1 (7.1)
**Nickel**	6	5 (83.3)	1 (16.7)	64.8 (±13.0)	5 (83.3)	-	1 (16.7)
**Formaldehyde**	33	25 (75.8)	8 (24.2)	62.0 (±11.9)	8 (24.2)	16 (48.5)	9 (27.3)
**Textile dust**	85	40 (47.1)	45 (52.9)	68.5 (±10.1)	34 (40.0)	45 (52.9)	6 (7.1)
**Total**	240	155	85		145	67	28

Note: the number of exposures can be equal or higher than the number of exposed subjects because more than one work period is considered in the exposure history for each SNC case.

**Table 2 ijerph-18-12593-t002:** Occurrences (*n*) of exposure (or tasks) to an IARC carcinogenic agent in jobs for the nasal cavity and paranasal sinuses in atypical working situations assessed by industrial hygienists of each COR.

Carcinogenic Agent	Job (or Task Performed)	*n*
**Wood dust**		
Wood dusts (Generic)	Construction-related jobs (architect, surveyor, bricklayer, carpenter, house painter, parquetry layer)	17
	Framer	4
	Production of resins (use of wooden shavings among raw materials)	2
	Wooden mold production in the steel industry	2
	Wood sculptor	1
	Production of wooden pallets	1
	Cooper (*wine*)	1
	Production of wooden buttons	1
	Pipe organ and keyboard assembly	1
	Production of table tennis equipment	1
	Manufacture of clogs	1
Straw	Binding up glass vessels in straw	2
	Binding up chairs in straw	1
	Straw pressing	1
Straw and sawdust	Agriculture and livestock	7
Sawdust	Poultry breeding (use of saw litters)	14
	Cork production	6
	Heel factory worker	5
	Cork grinding	2
	Cork processing	2
	Footwear finishing and miller	2
	Metalworker	1
	Butcher	1
	Printer	1
	Loading dock in harbor	1
	Consultant in raw cork selection	1
Leather dust	Office employee (going to production sector)	1
	Gluer and leather miller for heel covering	1
	Upper edger in leather and synthetic shoe and bag production	1
Chromium	Pipe and gas pipeline welding in urban and industrial facilities	2
	Use of chrome paints for motor vehicles	1
	Designer (use of chromic acid for development)	1
	Photoengraver (use of hexavalent chromium, Cr(VI))	1
	Chrome plating in the manufacture of housewares	1
	Chrome plating of rollers for punch cards	1
Formaldehyde	Hairdresser, barber	7
	Pastry chef	4
	Baker	4
	Tire production finishing	3
	Plastic production	3
	Plastic printer	2
	Cap production in cork mill	1
	Spinning employee at wool mill	1
	Abrasive material processing	1
	Extractor hood production	1
	Iron foundries	1
	Additive production for pharmacological use (chemical industry)	1
	Clutch and brake lining production	1
Textile dust	Dressmaker	18
	Technical assistant/mechanical textile worker	14
	Wool and artificial fiber spinning	12
	Knitter	5
	Upholsterer	5
	Other textile job titles (sorting rags, fraying of fabric, spool making, stocking production)	5
	Clothing and fabric packer	4
	Loom operator	4
	Fabric cutting	4
	Weaver	3
	Textile company delivery man	1
	Gallery ticket holder	1
	Department head (manufacturing, cotton mill)	1
	Contract apprentice	1
	Trader	1
	Ironer	1
	Starch helper	1
	Woodworker (furniture assembly)	1

**Table 3 ijerph-18-12593-t003:** Occurrences (n) of jobs (or tasks) with coexposures to IARC carcinogenic agents for nasal cavities and paranasal sinuses.

Coexposure to Carcinogenic Agents	Job (or Task Performed)	*n*
Wood and leather dusts	Lapin fur stitcher (use of sawdust for drying between fur and leather) Rabbit skin cleaner (supposed use of sawdust for drying between fur and leather) Boy from fur shop (supposed use of sawdust for drying between fur and leather) Processing of heels	1 1 1 1
Wood and textile dusts	Layer and painter of wooden floors and carpets and plastic floor installer	1
Wood dust and chromium	Engineering laboratory activities	1
Wood dust and welding fumes	Coffin production at funeral home	1
Wood dust and formaldehyde	Livestock farming (formaldehyde)	1
Cork and leather dust	Cork sole production	1
	Cork sole scratcher in footwear (use of solvents)	1
Leather dust and nickel	Gold jewelry cleaner using skins	1
Leather dust and chromium	Maker of models and seller of leather garments, traveler at tanneries in Turkey	1
Textile dusts and formaldehyde	Loom operator in woolen mill and spinning	1
	Apprentice at a tailor’s shop	1
Chromium and nickel	Graphic designer	1
	Nickel and chrome paint producer	1
	Steel industry, chrome and nickel addition to obtain special steels	1
	Chrome use in anodic oxidation of aluminum	1
	Use of chromic acid for degreasing and use of chrome and nickel paints	1

## Data Availability

Data sharing not applicable (restrictions apply to the availability of these data).

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
