# Peer review of "Exposures to IARC Carcinogenic Agents in Work Settings Not Traditionally Associated with Sinonasal Cancer Risk: The Experience of the Italian National Sinonasal Cancer Registry"

_ijerph, 2021, doi:10.3390/ijerph182312593_

Round 1

Reviewer 1 Report

The authors presented occupational exposures to carcinogenic agents that have not been conventionally considered as high-risk factors of SNC. They tried to provide and summarize relevant evidence using 216 histologic-verified SNC cases from ReNaTuNS.

My biggest concern is that the sample size is too small to generalize the authors’ discussion. To be specific, one or two instances of occupation exposed to carcinogen agents is a too small number to support their association. For example, the data has one butcher who has been exposed to sawdust, which is not necessarily suggesting that common butchers tend to be exposed to sawdust and to develop SNC. I could find many similar statements for other occupations that were based on a single observation in the Discussion. I would suggest categorizing occupations in a broad way to generalize the association.

Another concern of mine is the data quality of exposure. I conjecture that the authors wanted to describe it with three categories by ‘probability of exposure’ in Table 1. However, I cannot find where the authors used this information to support their findings.

Figure 1 for the distribution of SNC morphology can be tricky for a fair comparison between exposures because the number of observations is different in each exposure; for example, about 70% of adenocarcinoma exposed to wood dust were based on 89 people but 50% of adenocarcinoma exposed to nicker was based on only 6 people. It can be misleading if the data summary does not include any information about the uncertainty that depends on sample size. 

Tables 2&3 could have been better summarized rather than showing all details, which may make readers overwhelmed. As mentioned earlier, I suggest making broader categories of occupations that potentially can provide a clear idea of considering the type of jobs in the analysis of SNC.

Overall, it can be a good question to summarize SNC cases with a more extensive perspective of occupational exposures aiming at detecting non-traditional jobs with a high risk of SNC. However, I think the authors’ discussion is a hasty generalization due to the lack of data and the disorganized summaries.

Author Response

Reviewer 1: Comments and Suggestions for Authors

The authors presented occupational exposures to carcinogenic agents that have not been conventionally considered as high-risk factors of SNC. They tried to provide and summarize relevant evidence using 216 histologic-verified SNC cases from ReNaTuNS.

  • My biggest concern is that the sample size is too small to generalize the authors’ discussion. To be specific, one or two instances of occupation exposed to carcinogen agents is a too small number to support their association. For example, the data has one butcher who has been exposed to sawdust, which is not necessarily suggesting that common butchers tend to be exposed to sawdust and to develop SNC. I could find many similar statements for other occupations that were based on a single observation in the Discussion. I would suggest categorizing occupations in a broad way to generalize the association.

Answer:

We agree with the reviewer that our results on the presence of exposure to a certain agent in a job or a sector are not generalizable to the working population. However, the study should be intended as a large case series, whose aim was rather to provide the readers with a catalogue of situations in which SNC cases have been exposed to carcinogens known or suspected for causing these tumours in sectors and jobs where such an exposure would have not been expected. In our view, this work may be relevant in guiding public health practitioners in the identification of occupational etiological agents of these tumours, suggesting the possible existence of these carcinogens in unusual sectors and jobs. We added a short paragraph on this issue in the discussion.

  • Another concern of mine is the data quality of exposure. I conjecture that the authors wanted to describe it with three categories by ‘probability of exposure’ in Table 1. However, I cannot find where the authors used this information to support their findings.

Answer: in the Methods we have added supplemental information about the classification of exposure

  • Figure 1 for the distribution of SNC morphology can be tricky for a fair comparison between exposures because the number of observations is different in each exposure; for example, about 70% of adenocarcinoma exposed to wood dust were based on 89 people but 50% of adenocarcinoma exposed to nicker was based on only 6 people. It can be misleading if the data summary does not include any information about the uncertainty that depends on sample size.

Answer: we consider that this figure is not misleading, being a distribution of histologic types by exposures to specific carcinogenic agents. Anyway, for a better comprehension, we have substituted the figure with counts instead of percentages.

  • Tables 2&3 could have been better summarized rather than showing all details, which may make readers overwhelmed. As mentioned earlier, I suggest making broader categories of occupations that potentially can provide a clear idea of considering the type of jobs in the analysis of SNC.

Answer:

Given the nature of case series of this study, we actually believe that the detailed description of the job titles held by cases gives a better understanding to the reader regarding the variety of jobs and activities where exposure to each of the carcinogens considered may have occurred.

  • Overall, it can be a good question to summarize SNC cases with a more extensive perspective of occupational exposures aiming at detecting non-traditional jobs with a high risk of SNC. However, I think the authors’ discussion is a hasty generalization due to the lack of data and the disorganized summaries.

Answer:

We do not agree about the supposed generalization of the discussion, due to the lack of data and reporting confused summaries. Rather, we reported the findings of the scientific literature about occupational exposures to the selected IARC carcinogens exactly with the aim of suggesting hypotheses during the investigation of a SNC case who had worked in a sector not recognized at risk for SNC. We extended this point in the discussion, and outlined the epidemiological implications in terms of preventive measures to be implemented and insurance claims for SNC patients.

Reviewer 2 Report

Thank you for undertaking this work, it is an interesting manuscript and may be quite valuable across a number of users. Some aspects of the paper, particularly the methodological description and conclusions drawn, require editing to provide clear explanations and thus value to the reader.

Introduction

Lines 67-72: Tobacco smoking should not be included in this list. Please also specify the cancer sites referred to (Nasal cavity and paranasal sinus).

Lines 88-89: As a non-Italian it is not clear to me what “national territory” refers to. I would prefer to see reference to proportion of the population covered.

Lines 92-93: I think you mean “…may ultimately lead to the implementation of preventive measures”.

Materials and Methods

This section could use some revision to clarify for the reader how information passes into the CORs. What proportion of SNCs are captured, and what proportion of those are assessed for occupational exposure via questionnaire?

Lines 124-126: Again, please specify the cancer sites referred to.

As currently written, the reader cannot determine how “working settings not traditionally associated with SNC risk” were defined. More detail is needed.

Methods used to determine “probability of exposure” need to be described.

Results

Line 128: “CORs selected 216 out of 2,208 SNC cases recorded in the ReNaTuNS database” – clarify what is meant here.

Lines 157-158: The findings reported here are based on self-reported exposures obtained through questionnaire. The authors cannot be certain that exposures “occurred”, and exposures were certainly not “observed”. Please use language throughout the paper that makes this clear.

Line 169: Is “knitwears” an occupation?

Table 2: Again, please specify the cancer sites referred to. Also, “metal furniture” and “knitwear” is not a job nor a task. Please review this table to ensure all are clearly defined.

Line 173: Please change “causal agents” to “carcinogenic agents” and define which groups of IARC agents (e.g., 1, 2A, etc.) you are referring to.

Are you referring to hexavalent chromium in the results? If yes, this needs to be specified. If no, more detail is needed to explain the limitation that you are not able to differentiate it.  

Discussion

There are many English mistakes in this section, please review and edit.

Some of the agents summarized are simply a re-iteration of results.

Lines 200-205: This simply reads as a summary of results. Where is the context with respect to other studies?

Lines 333-346: The decision to describe morphology seems to come out of nowhere - this should be introduced in prior section(s)

Lines 348-364: Please identify this as a subsection focused on strengths and limitations of the work. More detail is needed as well, for instance what is the comprehensiveness of these findings in terms of identifying jobs and tasks not commonly associated with SNCs in Italy? And, how might these findings translate to other countries/the global research community? So much reference is made to IARC, yet the authors do not highlight potential implications for identifying new research targets that could inform future Monographs. 

In addition to clearly defined strengths and limitations, I would like to see the authors explain how this work is informing compensation policy in Italy (this is not clear at all) and next steps in terms of research needs, prevention measures that could be undertaken.

Author Response

Reviewer 2: Comments and Suggestions for Authors

Thank you for undertaking this work, it is an interesting manuscript and may be quite valuable across a number of users. Some aspects of the paper, particularly the methodological description and conclusions drawn, require editing to provide clear explanations and thus value to the reader.

Introduction

  • Lines 67-72: Tobacco smoking should not be included in this list. Please also specify the cancer sites referred to (Nasal cavity and paranasal sinus).

Answer: tobacco smoking is included in IARC group 1 substances (with sufficient evidence of carcinogenicity in humans for nasal cavity and paranasal sinus), see reference n.5. We have now specified along the text the cancer sites with reference to nasal cavity and paranasal sinus as target organs.

  • Lines 88-89: As a non-Italian it is not clear to me what “national territory” refers to. I would prefer to see reference to proportion of the population covered.

Answer: we have explained such information in the Methods

  • Lines 92-93: I think you mean “…may ultimately lead to the implementation of preventive measures”.

Answer: we have corrected according to the Reviewer

Materials and Methods

  • This section could use some revision to clarify for the reader how information passes into the CORs. What proportion of SNCs are captured, and what proportion of those are assessed for occupational exposure via questionnaire?

Answer: we have clarified such information in the Discussion

  • Lines 124-126: Again, please specify the cancer sites referred to.

Answer: We have specified the cancer sites with reference to nasal cavity and paranasal sinus as target organs

  • As currently written, the reader cannot determine how “working settings not traditionally associated with SNC risk” were defined. More detail is needed.

Answer: more details are provided, but in the Results

  • Methods used to determine “probability of exposure” need to be described.

Answer: we have added supplemental information about the classification of exposure

Results

  • Line 128: “CORs selected 216 out of 2,208 SNC cases recorded in the ReNaTuNS database” – clarify what is meant here.

Answer: the selection criteria of the 216 SNC cases have been explained more in details

  • Lines 157-158: The findings reported here are based on self-reported exposures obtained through questionnaire. The authors cannot be certain that exposures “occurred”, and exposures were certainly not “observed”. Please use language throughout the paper that makes this clear.

Answer: the language was modified accordingly

  • Line 169: Is “knitwears” an occupation?

Answer: it has been corrected in “Knitters”

  • Table 2: Again, please specify the cancer sites referred to. Also, “metal furniture” and “knitwear” is not a job nor a task. Please review this table to ensure all are clearly defined.

Answer: we have specified both cancer sites and the jobs (“metalworker” and “knitter”)

  • Line 173: Please change “causal agents” to “carcinogenic agents” and define which groups of IARC agents (e.g., 1, 2A, etc.) you are referring to.

Answer: we have made the requested changes

  • Are you referring to hexavalent chromium in the results? If yes, this needs to be specified. If no, more detail is needed to explain the limitation that you are not able to differentiate it.

Answer: in the text ‘chromium’ is referred to the chemical element, ‘chrome’ to plating or compounds

Discussion

  • There are many English mistakes in this section, please review and edit.

Answer:

Thanks for noting English mistakes. We have now thoroughly revised the English language throughout the manuscript.

  • Some of the agents summarized are simply a re-iteration of results.

Answer: we describe the occupational exposures to the selected IARC carcinogens reported in the scientific literature to suggest hypotheses when investigating the occupational history of a SNC case who had worked in a sector not recognized at risk for the sinonasal cancer. From an epidemiological point of view, this is an important tool for implementing preventive measures and supporting insurance claims for SNC patients.  All these items are now implemented in the discussion

  • Lines 200-205: This simply reads as a summary of results. Where is the context with respect to other studies?

Answer:

We added the following sentence in response to this reviewer’s comment, referring to the results of a meta-analysis (also conducted by our research group) to summarize the evidence on the risk of SNC in the jobs described in the paragraph, all belonging to the construction industry. “A meta-analysis estimated a 62% excess risk of SNC for previous work in the construction industry, and wood dust appears the agent mainly responsible for this association (Binazzi et al., 2015)”

  • Lines 333-346: The decision to describe morphology seems to come out of nowhere - this should be introduced in prior section(s)

Answer: we introduced a preliminary description of histologic types in the methods in the Methods

  • Lines 348-364: Please identify this as a subsection focused on strengths and limitations of the work. More detail is needed as well, for instance what is the comprehensiveness of these findings in terms of identifying jobs and tasks not commonly associated with SNCs in Italy? And, how might these findings translate to other countries/the global research community? So much reference is made to IARC, yet the authors do not highlight potential implications for identifying new research targets that could inform future Monographs.

Answer: we have implemented the requested issues in the text

  • In addition to clearly defined strengths and limitations, I would like to see the authors explain how this work is informing compensation policy in Italy (this is not clear at all) and next steps in terms of research needs, prevention measures that could be undertaken.

Answer: more details have been implemented, with regard to the issues of compensation and preventive measures.

Round 2

Reviewer 1 Report

The authors presented occupational exposures to carcinogenic agents that have not been conventionally considered as high-risk factors of SNC. They tried to provide and summarize relevant evidence using 216 histologic-verified SNC cases from ReNaTuNS.

My biggest concern is that the sample size is too small to generalize the authors’ discussion. To be specific, one or two instances of occupation exposed to carcinogen agent is too small number of evidence to support their association. For example, the data has one butcher who has been exposed to sawdust, which is not necessarily suggesting that common butchers tend to be exposed to sawdust and to develop SNC. I could find many of similar statements for other occupations based on a single observation in the Discussion. I would suggest categorizing occupations into a broad way to generalize the association.

Another concern of mine is the data quality of exposure. I conjecture that the authors wanted to describe the quality of exposure with three categories by ‘probability of exposure’ in Table 1. However, I cannot find where the authors used this information to support their findings.

Figure 1 for the distribution of SNC morphology can be tricky for fair comparison between exposures because the number of observations is different in each exposure; for example, about 70% of adenocarcinoma exposed to wood dusts were based on 89 people but 50% of adenocarcinoma exposed to nicker was based on only 6 people.

Table 2&3 could have been better summarized rather than showing all details, which may make readers overwhelmed. As mentioned earlier, I suggest to make broader categories of occupations that potentially can provide clear idea of considering type of jobs in the analysis of SNC.

Overall, it is a good question to summarize SNC cases with a more extensive perspective of occupational exposures aiming at detecting non-traditional jobs with high risk of SNC. However, I think the authors’ discussion may be hasty generalization mainly due to lack of data and possibly due to disorganized summaries.

====================================================

Second review comments

Overall, the authors addressed my comments satisfactorily. I appreciate that the authors edited the discussion that stressed more on the study aim of providing a catalog of unusual job exposure rather than generalizing an instance. Also, Figure 1 with the count unit makes more sense to me now.

Thank the authors for clarifying the classification of occupational exposure in the Section Method and Table 1. However, I would like to add two comments in this regard. First, use a consistent word either certain or definite. Secondly, the percentile (%) of Probability of exposure should be read correctly. Because the percentiles were based on the total of each of ‘definite’, ‘probable’ and ‘possible’, not on the total of each agent, the percentiles indicate the distribution of agents within each type of exposure and therefore, it is not appropriate to state like P4L170-173. The current percentiles are read such that ‘wood dust was the most frequent for definite’, which may not be the point that the authors wanted to highlight. Also in fact, for the formaldehyde, ‘probable’ (N=16) was the most frequent, not ‘possible’ (N=9). If it is anticipated to be read like ‘definite for the most agents’, or ‘probable was the most frequent for textile dust and for formaldehyde’, the percentile should be computed based on N of column All of Table 1. In this sense, it would be great if all other percentiles reported in the manuscript are double-checked.

Please check grammars overall, like broken sentences, wrong hyphens, no periods; e.g., L87-91 incomplete sentence, L98 docu-mentation, L103 mor-tality, L149 period, L252 construc-tion, L311 chro-mium, L375 it is noticeable to refer of…, L380 it could not be was not possible…, L401 fea-ture, L403 In-surance, L404 pa-tient’s (patients’?), L408 compen-sation.

Author Response

Reviewer 1:

Second review comments

Overall, the authors addressed my comments satisfactorily. I appreciate that the authors edited the discussion that stressed more on the study aim of providing a catalog of unusual job exposure rather than generalizing an instance. Also, Figure 1 with the count unit makes more sense to me now.

Thank the authors for clarifying the classification of occupational exposure in the Section Method and Table 1. However, I would like to add two comments in this regard. First, use a consistent word either certain or definite.

Answer: the word ‘certain’ has been replaced by ‘definite’

Secondly, the percentile (%) of Probability of exposure should be read correctly. Because the percentiles were based on the total of each of ‘definite’, ‘probable’ and ‘possible’, not on the total of each agent, the percentiles indicate the distribution of agents within each type of exposure and therefore, it is not appropriate to state like P4L170-173. The current percentiles are read such that ‘wood dust was the most frequent for definite’, which may not be the point that the authors wanted to highlight. Also in fact, for the formaldehyde, ‘probable’ (N=16) was the most frequent, not ‘possible’ (N=9). If it is anticipated to be read like ‘definite for the most agents’, or ‘probable was the most frequent for textile dust and for formaldehyde’, the percentile should be computed based on N of column All of Table 1. In this sense, it would be great if all other percentiles reported in the manuscript are double-checked.

Answer: all the percentages in table 1 are now computed by row (and not by column) and the corresponding text in the results has been modified accordingly.

Please check grammars overall, like broken sentences, wrong hyphens, no periods; e.g., L87-91 incomplete sentence, L98 docu-mentation, L103 mor-tality, L149 period, L252 construc-tion, L311 chro-mium, L375 it is noticeable to refer of…, L380 it could not be was not possible…, L401 fea-ture, L403 In-surance, L404 pa-tient’s (patients’?), L408 compen-sation.

Answer: we have checked and corrected the above issues
